# Gold Ion Beam Milled Gold Zero-Mode Waveguides

**DOI:** 10.3390/nano12101755

**Published:** 2022-05-21

**Authors:** Troy C. Messina, Bernadeta R. Srijanto, Charles Patrick Collier, Ivan I. Kravchenko, Christopher I. Richards

**Affiliations:** 1Department of Physics, Berea College, 101 Chestnut Street, Berea, KY 40404, USA; 2Center for Nanophase Materials Science, Oak Ridge National Labs, Oak Ridge, TN 37831, USA; srijantobr@ornl.gov (B.R.S.); colliercp@ornl.gov (C.P.C.); kravchenkoii@ornl.gov (I.I.K.); 3Department of Chemistry, University of Kentucky, 209 Chemistry-Physics Building, Lexington, KY 40202, USA; chris.richards@uky.edu

**Keywords:** single molecule spectroscopy, nanostructures, single molecule, sub-wavelength apertures, zero-mode waveguides

## Abstract

Zero-mode waveguides (ZMWs) are widely used in single molecule fluorescence microscopy for their enhancement of emitted light and the ability to study samples at physiological concentrations. ZMWs are typically produced using photo or electron beam lithography. We report a new method of ZMW production using focused ion beam (FIB) milling with gold ions. We demonstrate that ion-milled gold ZMWs with 200 nm apertures exhibit similar plasmon-enhanced fluorescence seen with ZMWs fabricated with traditional techniques such as electron beam lithography.

## 1. Introduction

Single molecule fluorescence spectroscopy (SMS) is widely used as an experimental technique to examine the real-time dynamics of biological systems with high spatial and temporal resolution [1,2,3,4]. Spatial resolution has been pushed to its limits with a variety of techniques such as laser scanning confocal microscopy and super-resolution techniques such as Stochastic Optical Reconstruction Microscopy (STORM) and Photoactivated Localization Microscopy (PALM) [5]. Both STORM and PALM rely on fluorescent molecules that switch between dark and emitting states so that overlapping diffraction-limited single molecule emission spots can be distinguished. The temporal resolution of single molecule studies is primarily limited by the photon flux of individual emitters. Non-radiative relaxation from the excited state reduces the fluorescence quantum yields of molecular probes used in fluorescence experiments, which affects the maximum rate at which information is obtained by detected photons. Amelioration comes from advances being made with techniques such as near-field scanning optical microscopes [6,7], the use of nanostructures such as nano-antennas [8,9,10,11,12,13], and zero-mode waveguides (ZMW) [14]. Nano-antennas and zero-mode waveguides enhance fluoresence by creating a surface plasmon resonance that is confined to an active region within the nanostructure. The surface plasmon electric field interacts with the electric dipole of a fluorescent molecule occupying the active region of the nanostructure. This interaction affects both the excitation and the emission of the fluorescent molecule [15]. Experimental dissection of the contributions to excitation and emission have been made [16]. A theory of spontaneous emission related to nanostructures and their substrates has also been developed [17]. Our interest in this work is with ZMW because of their potential to improve the temporal resolution in SMS.

Zero-mode waveguides are arrays of thousands to millions of sub-wavelength diameter holes, typically 50–250 nm, in a thin metallic film of 100–200 nm thickness. Having a diameter smaller than the wavelength of excitation light, the ZMW acts as a plasmonic resonator that may enhance fluorescence absorption and emission [18,19,20,21]. As a new approach in SMS, the characterization of ZMW related to size and shape [21,22,23,24,25], metal and layering compositions [18,21,22,26,27,28,29], excitation wavelength [19], spatial position of fluorophores relative to the metal structures [30,31,32], and spectral overlap of the surface plasmon resonance and excitation/emission of fluorophores [33,34,35,36,37] have been highly active areas of research. The geometry and opacity of the ZMW enables measurements at physiological concentrations (∼1–100 μM) in SMS [30,38,39,40,41,42,43,44,45,46]. As a result, ZMWs have found wide application in studies of genomic sequencing [47,48,49,50], protein–protein interaction [51,52,53,54], ligand–receptor binding [55,56,57,58,59,60,61], membrane-bound diffusion events [62,63] and the study of membrane proteins [64,65]. ZMWs have also been used to manipulate exciton behavior in quantum dots [66], and ZMWs have been shown to extend Förster Resonance Energy Transfer (FRET) distances from 10 to 13.6 nm [67]. A recent mini-review describes applications of these nanostructures for single molecule experiments [68].

A variety of ZMW fabrication techniques have been reported, including deep UV photolithography [69], electron beam lithography [14,18,26,69], colloidal lithography [70], and focused ion beam (FIB) milling with gallium ions [25,29,71,72]. Each of these techniques has advantages and disadvantages. Lithography can be used to create millions of ZMW holes in a relatively short period of time, but it suffers from resist contamination that is difficult to remove, has unknown interactions with single molecules being studied, and leads to autofluorescence that overwhelms single molecule fluorescence. Furthermore, difficulties during the lift-off process occur because the resist features have high aspect ratios and small lateral dimensions. This is particularly the case for small ZMW hole diameters <100 nm. Colloidal lithography is inexpensive, but it is more challenging to obtain regular feature sizes and shapes. FIB overcomes the disadvantages of the lithographic techniques but takes significantly longer to write an array of holes compared to lithography techniques, and the holes tend to have a tapered shape with a smaller diameter at the metal–glass interface. FIB has typically used gallium ions as the milling beam. Ion milling overcomes resist and feature regularity issues. The major drawback of using gallium ions is that it can implant into the metal film, changing the electronic structure and leading to variations in the plasmonic properties. This requires post-fabrication chemical processing using a gold etchant to remove the gallium-implanted layer. The etchant often alters the final ZMW hole diameter and shape, leading to heterogenous features with non-optimized properties. In this work, we introduce a new approach for fabricating ZMWs with FIB by utilizing double-ionized gold as the ion beam (Au${}^{++}$). The resulting ZMWs do not suffer from pattern irregularities or contamination, eliminating the needs for post-fabrication processing that could alter hole diameter and shape.

As a proof-of-concept, we fabricated gold ZMWs with gold ion milling. The ZMWs consisted of 33 × 33 arrays of 200 nm holes. The glass surface at the bottom of ZMW holes was functionalized and labeled with Atto647. Time-series trajectories were measured with single photon counting and compared to Atto647 bound to functionalized glass surfaces. Low concentration surface functionalization is a method that allows the direct comparison of non-ZMW and ZMW fluorescence. The functionalization methods we use are designed to ensure fluorescent molecules only attach in the ZMW holes, eliminating extraneous signals that would convolute the result. We also present an analysis technique that is capable of extracting photophysical states of dye molecules on glass and in ZMWs based on hidden Markov model analysis reported previously [73]. Fluorescence enhancements in gold-ion-milled ZMWs are comparable to the enhancements reported in ZMWs fabricated using lithography without the complications of contamination requiring difficult post-processing conditions.

## 2. Materials and Methods

### 2.1. Ion-Milled Zero Mode Waveguides

Gold zero-mode waveguides were made on 25 mm circular #1 coverslips. The coverslips were first cleaned by sonication at 50 °C for one hour in 5 M NaOH. The coverslips were rinsed in ultra pure water and subsequently sonicated at 50 °C for one hour in 0.1 N HCl. The coverslips were rinsed with ultra pure water followed by 100% ethanol and dried with flowing nitrogen.

The coverslips received a 100 nm coating of electron beam evaporated gold over a 5 nm electron beam evaporated adhesion layer of chromium. A Raith Velion focused ion beam/scanning electron microscope (FIB/SEM) was used to mill ZMW arrays. The arrays were designed using LayoutEditor and consisted of 33 × 33 milled 200 nm diameter holes. The gold-coated coverslips were ion milled with doubly ionized gold (Au${}^{++}$) at an energy of 35.2 kV at 2.1 pA and a total dose of 500 mC/cm${}^{2}$. The energy and dosage parameters were determined to provide good circular hole geometry and ensured holes were milled completely through the gold film to the glass surface. Milling completion was determined by monitoring the ion current for changes due to milling rate differences for gold, chromium, and glass. We recorded scanning electron micrographs (SEM) with a Zeiss Merlin FE-SEM. An example of a gold-ion-milled ZMW is shown in Figure 1. The holes are not perfectly circular as has been observed by others using FIB methods [72] and lithographic methods [71]. Improved circularity may be achieved by astigmatic adjustment of the focused ion beam.

### 2.2. Sample Preparation

Ion-milled ZMWs were prepared for single molecule spectroscopy by sonication at 50 °C for one hour in acetone followed by sonication at 50 °C for one hour in ethanol. The ZMWs were rinsed with ethanol and dried with flowing nitrogen.

Gold ZMWs of 200 nm diameter have been shown to provide the highest enhancements for fluorescent dyes in the red region of the visible spectrum [26]. We chose to use Atto647-NHS ester with an absorption maximum at $\lambda_{ex}=644$ nm and emission maximum at $\lambda_{em}=667$ nm. The dye was purchased from Sigma-Aldrich and used without modification. To bind Atto647 in the ZMWs, the glass surfaces were functionalized with (3-Aminopropyl)triethoxysilane (APTES) using 2% *v*/*v* in 100% ethanol. The ZMWs were soaked in APTES for one hour at room temperature. The ZMWs were rinsed with ethanol followed by ultra pure water. Before binding Atto647, the gold surfaces were passivated to prevent non-specific binding of the Atto dye. To passivate the gold, we use methoxy PEG Thiol, 2000 Da mPEG-SH, purchased from Laysan Bio and used without modification. The mPEG-SH is prepared at 20 μM concentration in 1× phosphate-buffered saline (PBS) pH 7.5 and was allowed to react with the gold surface for three hours at room temperature. The choice of 2 kD PEG was based on a recent study of passivation effects, where it was found that positively charged dyes such as Atto647 adhere to sample surfaces unless the surface is passivated with PEG longer than 0.5 kD and shorter than 5 kD [74]. The ZMW were rinsed with ultra pure water. Picomolar concentration Atto647-NHS was made in 1× PBS pH 8.5 and allowed to react with APTES on the ZMWs for one hour at room temperature. The orthogonal chemistry of silane-glass versus SH-gold ensures minimal non-specific binding, eliminating spurious background effects.

Control samples were made by cleaning glass coverslips using the base-then-acid method described above. Atto647 was bound to clean glass coverslips using the same procedures as described for ZMWs, skipping the the mPEG-SH passivation.

### 2.3. Single Molecule Spectroscopy

For fluorescence imaging, substrates (either ZMWs or glass) were placed onto a Mad City Laboratories piezo stage and raster scanned. An OBIS LX 640 nm 40 mW continuous beam laser was used for excitation after passing through a 60 × 1.45 NA oil immersion objective on an Olympus IX83 inverted microscope. Laser excitation was filtered with a Semrock 640/10 bandpass filter and set to 2 μW illumination power by a neutral density filter before entering the back aperture of the microscope. Excitation and emission were separated by a Semrock BrightLine^®^ multi-edge laser dichroic beamsplitter on the microscope. Emission was then passed through the microscope side port and a 100 nm pinhole for confocal imaging. A 650 nm long-pass dichroic mirror followed by a 673/44 bandpass were used to filter the detected emission. All dichroics and filters were purchased from Semrock. A Perkin-Elmer CD3226 avalanche photodiode was used to detect single photons, which were time-stamped by a PicoQuant PicoHarp 300. The microscope stage and PicoHarp were interfaced with a Windows PC running SymPhoTime 64 software.

Images of substrates were collected that were 5 × 5 or 10 × 10 μM${}^{2}$. When fluorescent molecules are observed in an image, the microscope stage is moved to the center of the fluorescent spot. Photon arrival times are collected from the molecule until a characteristic photobleaching step is observed and a background countrate (measured previously) is observed.

### 2.4. Methods of Analysis

Interphoton times (dt) were calculated from time-stamped photon arrivals at the detector. The reciprocal of this calculation (1/dt) provides the nearly instantaneous emission rate or intensity of the fluorescence. Each unique light-emitting state has an emission rate with a characteristic interphoton time described by an exponential decay assuming first-order kinetics between molecular light-emitting states. To determine these rates, the interphoton times were binned into a histogram and fit to two or three exponential term functions (Equation (Equation 1)). A decision between two or three exponential terms was made based on the reduced $\chi^{2}$, i.e., the quality of the fit (see Figure 2). Each exponential term represents background, fluorescent emitting state 1, and fluorescent emitting state 2.
(1)$$y\left(dt\right)=A_{1}e^{\left(-k_{1}dt\right)}+A_{2}e^{\left(-k_{2}dt\right)}\left(\cdots +A_{3}e^{\left(-k_{3}dt\right)}\right)$$

Each set of variables $A_{n}$ and $k_{n}$ represents a “hidden” molecular state because the different light-emitting states are unknown a priori. The variable $A_{n}$ represents the amplitude of the exponential function and relates to the fraction of time state *n* is active. This variable was not used in further analysis. The variable $k_{n}$ represents the characteristic emission rate in photons per second of state *n*. These $k_{n}$ are approximations to the emission rates. For a more accurate calculation of emission rates, $k_{n}$ values were used to construct a rate array of “hidden” molecular states in the photon trajectory. A Viterbi algorithm was applied to the data stream of interphoton times to reconstruct the pathway of molecular states [73,75]. Transition rates between states were set to extremely low values ($10^{-10}$ transitions per second) to prevent unlikely state-to-state transitions from un-optimized $k_{n}$ values. The $k_{n}$ values were adjusted to determine the best input rates based on maximizing the model likelihood calculated from the Viterbi algorithm. The Viterbi algorithm calculates the most likely molecular state for each detected photon. Thus, a molecular state pathway over time can be reconstructed on a photon-by-photon basis, which has the highest possible time resolution. A two-state reconstructed photon trajectory is shown in Figure 3, and examples of three-state trajectories are shown in Figure 4 for (a) glass and (b) ZMWs. Similar approaches using hidden Markov models and the Viterbi algorithm have been reported previously [60,73,75,76,77,78,79,80,81].

The molecular state reconstruction is used to collect the interphoton times associated with each molecular state. An average interphoton time is calculated, and its reciprocal gives the average countrate associated with each molecular state, $k_{n}^{\prime }$. The fluorescence emission intensity *I* is calculated as the difference between countrates $k^{\prime }$ of adjacent states, e.g.,
$$I_{2}=k_{2}^{\prime }-k_{1}^{\prime }$$
$$I_{1}=k_{1}^{\prime }-k_{0}^{\prime }$$

For example, in Figure 3, the first 6.01 s consists of 9769 photons corresponding to state 1 with an average interphoton time of $6.154\times 10^{-4}$ s. From 6.01 to 16.10 s consists of 6552 photons corresponding to state 0 with an average interphoton time of $1.541\times 10^{-3}$ s. The intensity of the fluorescent-emitting state is the difference between molecular state 1 and background molecular state 0 countrates, where we would calculate $I_{1}=k_{1}^{\prime }-k_{0}^{\prime }=1625-649=976$ photons/s. Figure 4 denotes the intensities for a three-state trajectory.

## 3. Results and Discussion

The fabrication quality of the ZMW arrays was verified using SEM. The holes milled using gold ions in gold films were highly consistent in shape. They were circular with a diameter of 200 nm. The fabrication was repeatable and of cleaner quality than ZMWs we fabricated using electron beam lithography.

Our fluorescence analysis includes 63 photon trajectories for Atto647 on glass and 87 photon trajectories for Atto647 in ZMWs. We observe that approximately one-half (33) of the trajectories on glass were best modeled by two states, a fluorescence-emitting and a background state. The remaining 30 are best described by three states, two fluorescence-emitting states and a background state. For Atto647 in ZMWs, approximately two-thirds (60 trajectories) are best modeled by two states, and 27 trajectories are three-state. A summary is given in Table 1.

The photon-by-photon trajectory reconstruction provides the opportunity to compare the different fluorescence-emitting states.

Figure 5 shows the probability distributions of the intensities measured for Atto647 on glass and ZMW substrates. Figure 5a is the intensity for trajectories modeled by a single fluorescing state such as that of Figure 3. Figure 5b,c are the intensity distributions for Atto647 with two fluorescing states, $I_{1}$ and $I_{2}$, respectively. Figure 5d is the intensity results for all $I_{1}$ and $I_{2}$ combined.

In all four cases, the ZMW intensity distributions are shifted toward higher count rates than those on glass. The enhancement factors (EF) calculated from averages of the distributions are 1.2, 1.8, 1.6, and 1.6 for Figure 5a–d, respectively. There were two intensity values on glass that were extremely high (greater than 16,000 photons per second). Ignoring these two potential outliers changes the enhancement factors to 1.6, 1.9, 1.6, and 1.8. This is comparable to what has been observed with Atto647 in 200 nm gold ZMWs previously [18,26]. Our enhancement factors are slightly lower for three possible reasons. (1) Ion milling is known to create conical cross-sections in holes, whereas lithographic techniques create a more vertical-walled hole. A smaller hole diameter would reduce the enhancement factor [21]. (2) Our analysis is based upon the photon stream, whereas past reports operated on binned trajectories [18,26], FCS [16,82], or FRET efficiencies [21]. Binning reduces resolution in a way that eliminates short-lived traversals that may contribute to the average intensity in a significant way. Much of the exponential probability density for molecular state lifetimes is at short interphoton times. (3) We have modeled many trajectories with three states, which reduces the countrate differences between states. For example, identifying the trajectory of Figure 4b as two states with a threshold between 4000 and 8000 photons per second (400 and 800 photons/100 ms) would artificially inflate the intensity, resulting in a larger ZMW enhancement factor.

The intensity distributions on glass are similar to one another for all states as seen in Figure 5a–c and in Table 1. For three-state trajectories on glass, there are clear photobleaching steps similar to the trajectory of Figure 4a. These are likely two Atto647 molecules in the confocal volume. Since the intensities of $I_{1}$ and $I_{2}$ are statistically similar, there is little quenching between the dye molecules, making the collection of all these states into a single distribution reasonable (Figure 5d). On the other hand, $I_{1}$ and $I_{2}$ in ZMWs are statistically distinguishable. The distribution for $I_{2}$ has approximately twice the intensity of $I_{1}$ in ZMWs, demonstrating that one-third of the molecules experience a very bright photophysical state that is unlikely due to two molecules in the confocal volume. Despite the statistical difference between $I_{1}$ and $I_{2}$ for ZMWs, we have combined the distributions for comparison in Figure 5d and in Table 1.

In our analysis, we limited the number of model states to two or three and use statistics on binned data to justify which model best fits. For three-state trajectories of glass substrates, there is evidence of two dye molecules in the confocal volume, since these trajectories tend to exhibit a double photobleaching step, such as the data shown in Figure 4a. The inclusion of these data do not greatly influence the resulting enhancement factor. The enhancement factor, however, is higher when including three-state ZMW trajectories than it is for two-state ZMW trajectories only. The ZMW three-state trajectories tended to exhibit blinking between two fluorescent emitting states such as that shown in Figure 4b rather than the photobleaching of Figure 4a. Using a threshold technique on histogram trajectories as in previous work would not distinguish these two distinct states, which would lead to higher enhancement factors than we report here [18,26].

## 4. Conclusions

Recent advances in focused ion beam milling enables the use of gold ions as the milling medium. We have demonstrated that milling zero-mode waveguides with doubly ionized gold results in devices of comparable characteristics to ZMWs made from photo or electron beam lithography. These 200 nm ion-milled ZMWs enhance fluorescence emission by a factor of approximately 2, which is comparable to previous studies. Furthermore, the fabrication of these ion-milled ZMWs is simpler, and these ZMWs do not suffer the contamination arising from lithographic resists, which are typically very bright emitters impairing single molecule fluorescence measurements.

## Figures and Tables

**Figure 1 nanomaterials-12-01755-f001:**
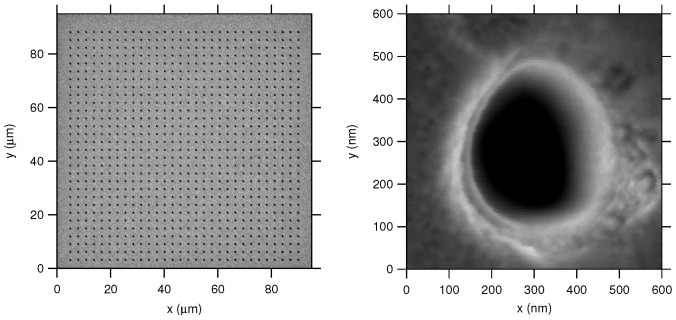
Scanning electron micrographs of ion-milled zero-mode waveguides. The ZMW consists of 200 nm holes in a 100 nm gold film. The ZMW array is 33 × 33.

**Figure 2 nanomaterials-12-01755-f002:**
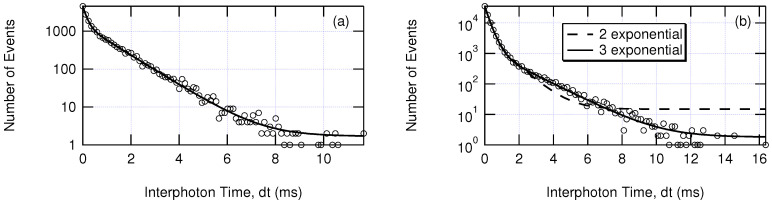
Multi-exponential fitting to histograms of interphoton times using Equation (Equation 1). (**a**) A photon trajectory well-fit by two exponential terms (background and fluorescent emitting states). (**b**) A photon trajectory better-fit by three exponential terms (background and two fluorescent emitting states).

**Figure 3 nanomaterials-12-01755-f003:**
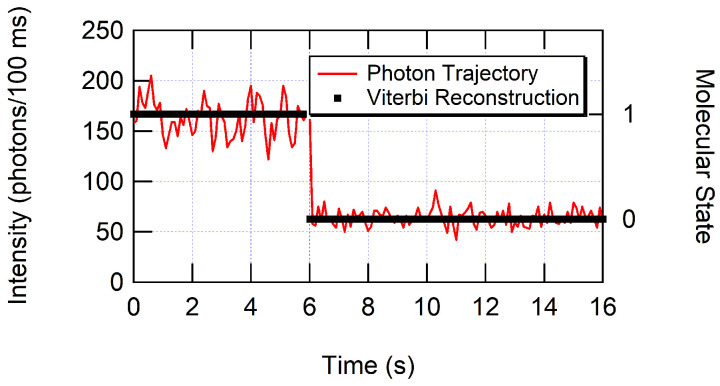
Photon arrival times are binned into a histogram with 100 ms bins to create the photon trajectory. A Viterbi algorithm is computed to determine the most likely molecular state pathway through the photon trajectory.

**Figure 4 nanomaterials-12-01755-f004:**
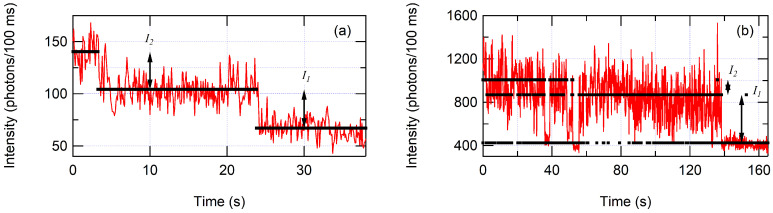
Three-state photon trajectories for (**a**) Atto647 on glass and (**b**) Atto647 in ZMWs. The overlaid Viterbi reconstruction is scaled to the average intensities. The intensities of Atto647 fluorescing states $I_{1}$ and $I_{2}$ are calculated as the difference between the average emission rates of adjacent states as indicated by arrows.

**Figure 5 nanomaterials-12-01755-f005:**
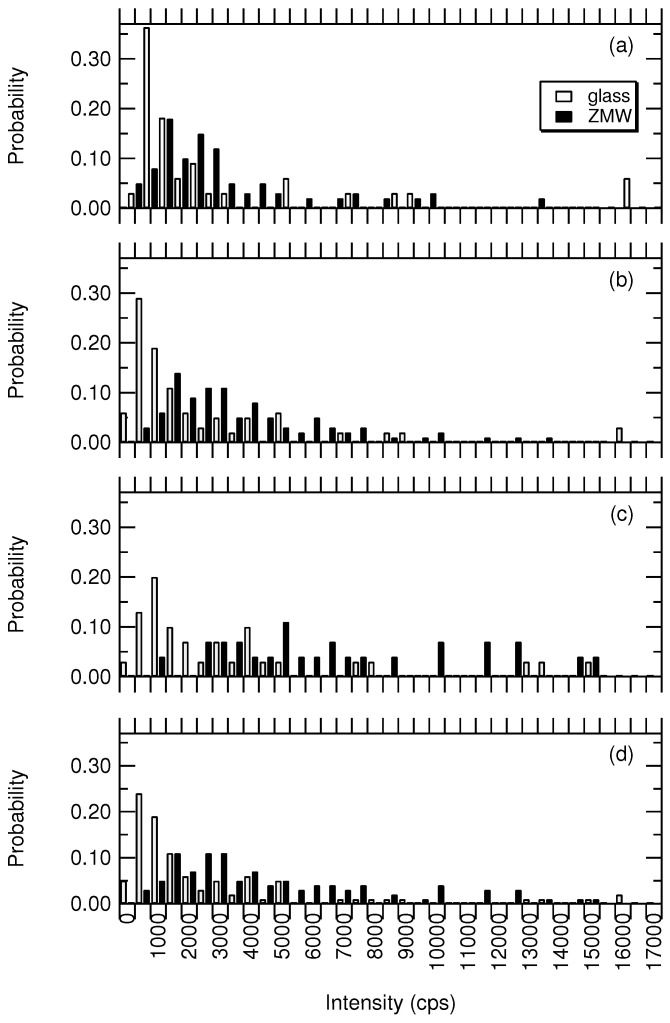
Probability distributions for the observed intensity of Atto647 on glass and in ZMWs for molecular states corresponding to (**a**) two-state trajectories, (**b**) state 1 in three-state trajectories, (**c**) state 2 in three-state trajectories, and (**d**) all states analyzed.

**Table 1 nanomaterials-12-01755-t001:** Fluorescence emission enhancements.

	Glass			ZMW			
**Distribution**	**Intensity**	**Std. Error**	**# Molecules**	**Intensity**	**Std. Error**	**# Molecules**	**EF**
Two-state, $I_{1}$	3086	722	33	3566	347	60	1.2
All, $I_{1}$	2572	401	63	4105	296	87	1.6
Three-state, $I_{2}$	3934	829	30	7224	752	27	1.8
All, $I_{1}$ and $I_{2}$	3011	384	93	4844	311	114	1.6

## Data Availability

The data presented in this study are available on request from the corresponding author.

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
