# Peer review of "Gold Ion Beam Milled Gold Zero-Mode Waveguides"

_nanomaterials, 2022, doi:10.3390/nano12101755_

Round 1

Reviewer 1 Report

The authors report a new method of focused ion beam (FIB) milling with gold ions beam for Zero-Mode Waveguides (ZMWs). Before publication, the authors should address following issues.

1. Suggest "Gold Ion Beam Milled Zero-Mode Waveguides" to "Gold Ion Beam Milled Gold Zero-Mode Waveguides"
2. check English expression.
3. define STORM, PALM
4. check "antennae" is correct expression.
5. add some comments of disadvantages for usage of gold ion instead of gallium ion.
6. better to add a comment for hole shape in Figure 1 (not a circular shape).
7. all company information should be consistently expressed.
8. critical issue, I suggest to add one more figure for actual experiment result of Single molecule fluorescence spectroscopy with this developed device.

Author Response

We write in response to the editor and reviewer comments for our manuscript “Gold Ion Beam Milled Zero-Mode Waveguides” (ID: nanomaterials-1722464) recently submitted to Nanomaterials. We appreciate the constructive feedback we received from you, the editor, and the reviewers who dedicated valuable time to respond to our work. We feel these reviews prompted revisions that result in a better manuscript. Below, we respond to the reviewer comments and indicate the revisions made to our manuscript. We hope these changes suitably address the expected revisions. We are re-submitting our revised manuscript with revisions highlighted in blue.

Sincerely,
Troy Messina

Reviewer 1:

The authors report a new method of focused ion beam (FIB) milling with gold ions beam for Zero-Mode Waveguides (ZMWs). Before publication, the authors should address following issues.

  1. Suggest "Gold Ion Beam Milled Zero-Mode Waveguides" to "Gold Ion Beam Milled Gold Zero-Mode Waveguides"
  • We have made this revision.
  1. check English expression.
  • We have edited the manuscript as required to improve grammar and readability.
  1. define STORM, PALM
  • We have added these definitions to the introduction section.
  1. check "antennae" is correct expression.
  • It is not clear if this comment is referring to antennae vs. antennas or the use of antennae for nanodevices. If the former, we have changed all instances of antennae to antennas. If the later, the expression nanoantenna has been used to describe a variety of plasmonic nanostructures in the literature previously, for example, our references [8], [9], and [10] as well as those listed below. We have added additional references.
    1. Song Sun, Ru Li, Mo Li, Qingguo Du, Ching Eng Png, and Ping Bai, “Hybrid Mushroom Nanoantenna for Fluorescence Enhancement by Matching the Stokes Shift of the Emitter,” Phys. Chem. C 122, 26, 14771–14780 (2018).
    2. M. Winkler, R. Regmi, V. Flauraud, J. Brugger, H. Rigneault, J. Wenger, M. F. García-Parajo, “Optical Antenna-based Fluorescence Correlation Spectroscopy to Probe the Nanoscale Dynamics of Biological Membranes,” J. Phys. Chem. Lett. 9, 110-119 (2018).
    3. Viktorija Glembockyte, Lennart Grabenhorst, Kateryna Trofymchuk, and Philip Tinnefeld, “DNA Origami Nanoantennas for Fluorescence Enhancement,” Chem. Res. 54, 17, 3338–3348 (2021).
  1. add some comments of disadvantages for usage of gold ion instead of gallium ion.
  • We have added statements about the tapering of FIB milled ZMWs and potential implanting of gold into the glass.
  1. better to add a comment for hole shape in Figure 1 (not a circular shape).
  • We have added statements about the non-circular shape. The lack of circularity has been observed in FIB and lithographic ZMWs. We show a much closer view of our ZMW than has been shown in other reports. This makes the shape appear more obviously non-circular.
  1. all company information should be consistently expressed.
  • We have rewritten portions of the Experimental section to express company information uniformly.
  1. critical issue, I suggest to add one more figure for actual experiment result of Single molecule fluorescence spectroscopy with this developed device.
  • The experimental data is the time of arrival of photons at the detector. This type of data is most often displayed as a histogram of photon arrival times binned into short time intervals over the total experimental collection time. Figures 3 and 4 show such single molecule trajectories binned at 100 ms.

Reviewer 2 Report

In this paper the authors discussed on the use of Au ion milling to prepare Zero Mode Waveguides (ZMWs) in Au film

The idea to prepare ZMWs by means of Focused Ion Beam (FIB) in not new and a significant number of papers have been reported (and cited in this paper). The use of Au ions to prepare the cavity is interesting and not reported so far (very few groups are using Au ions in the FIB system now)

The paper is rather well written, but there are several issues to be better discussed.

  1. the fabrication using FIB always leads to not vertical holes and it is also difficult to control the milling at the interface between the glass substrate and the metallic film. I recommend the authors to perform a FIB cross section to verify the shape of the hole.
  2. discussing the fluorescence enhancement in this way is very limiting. It is known that a circular hole in an Au film is not able to confine the field in a efficient way (for this reason alternative designs have been reported and used). The authors should discuss this aspect or eventually consider to perform a simulation
  3. a ZMW is typically used to perform single molecule detection at high concentration and not using molecule linked to the substrate. this is rather weird in my opinion. Anyway, the unknown profile of the e.m. field inside the hole makes it difficult to explain the observed signals.
  4. as largely demonstrated the FIB milling is critical at the interface between metal and glass, this could impact on the functionalization used here. for this reason an experiment where the molecules diffuse into the cavity is much better.
  5. the enhancement factor is very low. this could be due to the bad choice for the shape of the hole and the metal. An Aluminum ZMW could work better
  6. The used method of analysis does not enable time resolved measurements, FCS could give much more information on the fluorescence phenomane inside the holes

The authors mentioned a good number of references, anyway a recent review discussed several aspects related to single molecule spectroscopies in nanocavities, such as a ZMW. This paper could be also mentioned (Nanoscale Adv., 2021, 3, 633)

Author Response

We write in response to the editor and reviewer comments for our manuscript “Gold Ion Beam Milled Zero-Mode Waveguides” (ID: nanomaterials-1722464) recently submitted to Nanomaterials. We appreciate the constructive feedback we received from you, the editor, and the reviewers who dedicated valuable time to respond to our work. We feel these reviews prompted revisions that result in a better manuscript. Below, we respond to the reviewer comments and indicate the revisions made to our manuscript. We hope these changes suitably address the expected revisions. We are re-submitting our revised manuscript with revisions highlighted in blue.

Sincerely,
Troy Messina

Reviewer 2:

Comments and Suggestions for Authors

In this paper the authors discussed on the use of Au ion milling to prepare Zero Mode Waveguides (ZMWs) in Au film

The idea to prepare ZMWs by means of Focused Ion Beam (FIB) in not new and a significant number of papers have been reported (and cited in this paper). The use of Au ions to prepare the cavity is interesting and not reported so far (very few groups are using Au ions in the FIB system now)

The paper is rather well written, but there are several issues to be better discussed.

  1. the fabrication using FIB always leads to not vertical holes and it is also difficult to control the milling at the interface between the glass substrate and the metallic film. I recommend the authors to perform a FIB cross section to verify the shape of the hole.
    • We will investigate the ZMW hole shape as part of our continuing efforts to make these devices.
  2. discussing the fluorescence enhancement in this way is very limiting. It is known that a circular hole in an Au film is not able to confine the field in a efficient way (for this reason alternative designs have been reported and used). The authors should discuss this aspect or eventually consider to perform a simulation
    • We perform our analysis directly on the emitted photon stream. We feel this is the most direct method to interpret the fluorescence emission characteristics because we use the photon arrival times to calculate emission rates. Circular ZMWs have become commercially available, and there is currently very little commercial availability of other ZMW geometries. Therefore, we feel it is vital to fully understand the photophysics involved with circular ZMW geometries. Other plasmonic structures have the potential to provide higher levels of enhancement but they do not offer the same capability for confinement provided by the ZMW wells.
    • Our research group has reported simulations using FDTD in a previous publication for 200 nm aluminum ZMW, reference [64].
      1. Abdullah Al Masud, S. M. Nayeem Arefin, Fatema Fairooz, Xu Fu, Faruk Moonschi, Bernadeta R. Srijanto, Khaga Raj Neupane, Surya Aryal, Rosemary Calabro, Doo-Young Kim, C. Patrick Collier, Mustafa Habib Chowdhury, and Christopher I. Richards, “Photoluminescence Enhancement, Blinking Suppression, and Improved Biexciton Quantum Yield of Single Quantum Dots in Zero Mode Waveguides,” J. Phys. Chem. Lett. 12, 3303−3311 (2021).
    • Other groups have reported simulations of gold, aluminum, and layered aluminum/gold ZMWs of similar geometries as we report here including simulations due to tapering effects in the ion milling process, for example references [23] and [69]
      1. Meiyan Wu, Wenzhao Liu, Jinyong Hu, Zhensheng Zhong, Thitima Rujiralai, Lidan Zhou, Xinlun Cai, and Jie Ma, “Fluorescence enhancement in an over-etched gold zero-mode waveguide,” Optics Express 27, Issue 13, 19002-19018 (2019).
      2. P. Ponzellini, X. Zambrana-Puyalto, N. Maccaferri, L. Lanzanò, F. D. Angelis and D. Garoli, “Plasmonic zero mode waveguide for highly confined and enhanced fluorescence emission,” Nanoscale 10, 17362–500 (2018).
  1. a ZMW is typically used to perform single molecule detection at high concentration and not using molecule linked to the substrate. this is rather weird in my opinion. Anyway, the unknown profile of the e.m. field inside the hole makes it difficult to explain the observed signals.
    • We agree that one of the advantages of ZMWs is their applications to physiological concentration biological samples. The most widely used application for ZMWs and to our knowledge the only successful commercialization is through PacBio which relies on the immobilization of the fluorescently labeled nucleotides onto a polymerase. The immobilization of the fluorophore is what is used for the detection and identification of the base. For this reason, we believe it is important to understand the effect of the nanostructure on molecules that are immobilized to the bottom of the well. Our group has published several articles characterizing the photophysics of ZMWs using these surface functionalization techniques, see references [15], [24], [59] and [64]. Here, we are reporting similar ZMW characterization protocols to our group’s previously published work. FCS diffusion experiments are an indirect measure of fluorescence enhancement factors, whereas the analysis presented here uses the emitted photon stream directly.

      The EM field profiles in circular ZMWs has been extensively simulated in our cited references. These references include simulations of the tapering effects due to FIB milling methods.

  1. as largely demonstrated the FIB milling is critical at the interface between metal and glass, this could impact on the functionalization used here. for this reason an experiment where the molecules diffuse into the cavity is much better.
    • In our experience, low concentration surface functionalization is a method that allows direct comparison of non-ZMW and ZMW fluorescence. The functionalization methods we use are designed to ensure fluorescent molecules only attach in the ZMW holes, eliminating extraneous signals that would convolute the result. In FCS, the correlation function is fit to an assumed model. For a laser focused into solution a 3D gaussian is a valid model. This is not true for the excitation profile within a ZMW. Thus, a different excitation volume exists and needs to be accounted for between measurements in solution above a glass substrate and solution within the bottom of a ZMW. Additionally, diffusion experiments are performed at vastly different concentrations in non-ZMW and ZMW experiments, obfuscating whether the non-ZMW control experiment is applicable.
  2. the enhancement factor is very low. this could be due to the bad choice for the shape of the hole and the metal. An Aluminum ZMW could work better.
    • Our group has previously shown gold atto647 fluorophores are maximally enhanced in 200 nm diameter gold ZMWs. The enhancement factors we report are in good agreement with these previous publications. Higher enhancement factors have been reported using FCS, which are likely amplified by the fact that control experiments cannot be performed on solutions of the same high concentration as those in ZMWs. FCS does not directly measure the fluorescence intensity of single emitters; rather calculates the single emitter intensity from the correlation function based on several underlying model assumptions. Our analysis methodology directly measures the photon emission rates of single molecules on glass and in ZMW such that the comparison between the two scenarios is direct. The major emphasis of this work is demonstrating the utility of Au based FIB.
  3. The used method of analysis does not enable time resolved measurements, FCS could give much more information on the fluorescence phenomane inside the holes
    • Time resolved measurements have traditionally referred to measurements of fluorescence lifetimes. FCS on its own is not a time-resolved technique. It is an autocorrelation of the time-resolved photon or intensity stream. Here, we are analyzing the time-resolved photon stream (macro arrival times, not lifetime) directly, which gives both the highest possible time resolution, the time evolution, and a direct comparison of the fluorescence intensity of single emitters in different experimental scenarios. We agree that lifetime measurements are useful in determining the origin of fluorescence enhancement. However, the major emphasis of this work was to demonstrate that Au FIB can be used to generate ZMWs that can be used for single molecule experiments.
  4. The authors mentioned a good number of references, anyway a recent review discussed several aspects related to single molecule spectroscopies in nanocavities, such as a ZMW. This paper could be also mentioned (Nanoscale Adv., 2021, 3, 633)
    • Thank you for this reference, it has been added to our citations.

Round 2

Reviewer 1 Report

I recommend to be published.

Reviewer 2 Report

I think that the authors' reply and the improvements in the manuscript are good enough to enable the publication